# A Trip Back Home: Resistance to Herbivores of Native and Non-Native Plant Populations of *Datura stramonium*

**DOI:** 10.3390/plants13010131

**Published:** 2024-01-02

**Authors:** Juan Núñez-Farfán, Sabina Velázquez-Márquez, Jesús R. Torres-García, Ivan M. De-la-Cruz, Juan Arroyo, Pedro L. Valverde, César M. Flores-Ortiz, Luis B. Hernández-Portilla, Diana E. López-Cobos, Javier D. Matías

**Affiliations:** 1Departamento de Ecología Evolutiva, Instituto de Ecología, Universidad Nacional Autónoma de México (UNAM), Mexico City 04510, Mexico; svelazquez@ecologia.unam.mx (S.V.-M.); jrtorresg@ipn.mx (J.R.T.-G.); ivan.de.la.cruz.arguello@slu.se (I.M.D.-l.-C.); dlopezcobos@gmail.com (D.E.L.-C.); javier.matias@ecologia.unam.mx (J.D.M.); 2Departamento de Biología Vegetal y Ecología, Universidad de Sevilla, 41080 Sevilla, Spain; arroyo@us.es; 3Departament of Biology, Universidad Autónoma Metropolitana-Iztapalapa, Mexico City 09310, Mexico; plvp@xanum.uam.mx; 4Plant Physiology Laboratory, UBIPRO, FES Iztacala, Universidad Nacional Autónoma de México, Tlalnepantla 54090, Estado de Mexico, Mexico; cmflores@unam.mx (C.M.F.-O.); lbarbo@unam.mx (L.B.H.-P.)

**Keywords:** *Datura stramonium*, native and non-native populations, invasive species, tobacco flea beetle, datura striped beetle, tobacco weevil, enemy release hypothesis, increased competitive ability, tropane alkaloids, jimsonweed, toloache

## Abstract

When colonizing new ranges, plant populations may benefit from the absence of the checks imposed by the enemies, herbivores, and pathogens that regulated their numbers in their original range. Therefore, rates of plant damage or infestation by natural enemies are expected to be lower in the new range. Exposing both non-native and native plant populations in the native range, where native herbivores are present, can be used to test whether resistance mechanisms have diverged between populations. *Datura stramonium* is native to the Americas but widely distributed in Spain, where populations show lower herbivore damage than populations in the native range. We established experiments in two localities in the native range (Mexico), exposing two native and two non-native *D. stramonium* populations to natural herbivores. Plant performance differed between the localities, as did the abundance of the main specialist herbivore, *Lema daturaphila*. In Teotihuacán, where *L. daturaphila* is common, native plants had significantly more adult beetles and herbivore damage than non-native plants. The degree of infestation by the specialist seed predator *Trichobaris soror* differed among populations and between sites, but the native Ticumán population always had the lowest level of infestation. The Ticumán population also had the highest concentration of the alkaloid scopolamine. Scopolamine was negatively related to the number of eggs deposited by *L. daturaphila* in Teotihuacán. There was among-family variation in herbivore damage (resistance), alkaloid content (scopolamine), and infestation by *L. daturaphila* and *T. soror*, indicating genetic variation and potential for further evolution. Although native and non-native *D. stramonium* populations have not yet diverged in plant resistance/constitutive defense, the differences between ranges (and the two experimental sites) in the type and abundance of herbivores suggest that further research is needed on the role of resource availability and adaptive plasticity, specialized metabolites (induced, constitutive), and the relationship between genealogical origin and plant defense in both ranges.

## 1. Introduction

Release from natural enemies in a new range can promote a cascade of ecological and evolutionary changes at the species and community levels. The enemy release (ER) hypothesis predicts that plant and animal populations in a new range are freed from enemies, making them likely to establish and expand [1,2,3,4]. However, numerous studies, including meta-analyses of the predictions of the ER hypothesis, have reported mixed results (e.g., [5,6,7]). Non-native species in an introducing range may establish and expand depending on the effects of the enemy, how well-defended the host is, and the costs of defense. As a result, regulatory release and/or compensatory release can occur [4]. Also, the conclusions reached when comparing enemy-induced damage to non-native plants in an introduced range may differ between ‘biogeographical’ and ‘community’ comparisons [4,5,7,8]. At the community level, success in establishment in a new range is favored if non-natives possess higher resistance than native species in the same environment [2,9]. A recent analysis found that this expectation is met for woody plant species but not for non-woody plants [9]; further studies are needed to explain why non-woody plant species do not behave according to the expectation.

It has been proposed that the diversity of plant enemies, their impact on plants, and host plant adaptation are three factors that, acting in different ecological contexts, allow more precise predictions of the ecological and evolutionary outcomes of ER in the performance of non-native plants [7]. Thus, it is relevant to measure the (i) impact of enemies, (ii) their diversity (abundance and type of tissue consumed), and (iii) host adaptation (e.g., changes in the investment/type of defense and/or tolerance), to better predict the outcome of ER during the establishment of non-native plants [7]. 

A related question is whether ER spurs evolutionary divergence in resistance and/or tolerance to natural enemies of non-natives in the new range [2,4,7,10,11,12]. When non-native species possess strong but costly defenses, enemy release is expected to promote selection against genotypes with costly defenses, favoring reallocation of resources, for instance, to increase competitive ability (EICA; [13,14]). It is also possible that plants modify their original defenses to face the new enemies encountered in the introduced range [4,14]. Ecological genetics studies are particularly interested in population-level divergence in plant resistance. In the present study, we compared the damage caused by specialist and generalist insect herbivores to plants from native and non-native populations of the same species in the native habitat. Although it is more common to compare native to non-native populations in the introduced range (i.e., community [8]), performing the comparison in the native range provides a test of divergence between native and non-native populations.

The annual plant *Datura stramonium* is an excellent system to test potential evolutionary changes in plant resistance to herbivorous insects in non-native ranges after prolonged isolation from its original home range. Previous evidence strongly demonstrates that herbivore damage decreases plant fitness in the native home range [15,16,17,18,19,20]. Selection on resistance and tolerance varies depending on the herbivore species [18,20,21], whether it is a specialist or generalist, and whether it preys upon leaves or seeds [15,20,21,22]. 

Historical accounts indicate that *D. stramonium* was introduced from Mexico into Europe, particularly to southern Spain, during the XVI century, and it is currently a widespread and invasive plant there [23]. A previous biogeographical comparison of the levels of herbivory in different *D. stramonium* populations in Spain indicated that the average proportion of leaf area consumed by herbivores is much lower than the levels of damage recorded in native populations of this species [19,24]. Thus, at the biogeographical scale, *D. stramonium* is enemy released. This can result from the absence of native enemies in the non-native range (J. Arroyo, J. Núñez-Farfán and P.L. Valverde, personal observations), although local generalist insects have been recorded on plants of *D. stramonium* in Spain [24]. Furthermore, non-native populations have been reported to have lower concentrations of atropine and scopolamine and less herbivore damage than native populations [25]. Thus, the biogeographical comparison between populations of *D. stramonium* raises the question of whether the absence of plants’ natural enemies in the new range (Spain) has promoted differentiation in plant resistance traits relative to their native conspecifics. 

In this study, we aimed to perform a community comparison in the native range where the native specialist and generalist herbivores of *D. stramonium* co-occur. We assessed whether non-native plant populations of *D. stramonium* from Spain (a) have diverged in plant resistance from native populations (Mexico) and (b) differ in their production of the two major tropane alkaloids. Additionally, (c) we investigated whether the natural herbivores consume and/or oviposit more frequently on non-native plants. We expected that in the native habitat and in the presence of native specialized herbivores, non-native populations would have more damage and express lower levels of defensive alkaloids than native populations. To test this, we established two common garden experiments in two localities of central Mexico to grow plants of *D. stramonium* from the two ranges (Mexico and Spain) and exposed them to their natural herbivores. 

## 2. Results

### 2.1. Herbivores

The community of herbivorous insects consuming plants of *D. stramonium* was similar between the two experimental sites, with few exceptions (Figure 1). In both experimental sites, plants of *D. stramonium* were consumed by dietary specialists—the tobacco flea beetle (*Epitrix parvula*) and the striped datura beetle or three-lined potato beetle (*Lema daturaphila*, Coleoptera: Chrysomelidae). The generalist herbivore, the Chapulín grasshopper *Sphenarium purpurascens* (Orthoptera: Pyrgomorphidae) and the seed predator, the tobacco weevil (*Trichobaris soror*) (Coleoptera: Curculionidae), were also recorded. 

There were no significant differences in the number of *E. parvula* per plant between ranges (native *versus* non-native), among populations (Teotihuacán and Ticumán from Mexico; Valdeflores and Zubia from Spain) or among families (Figure 2A,B; Appendix A). However, there were differences among populations in the number of the seed predator *T. soror* in the two localities (Appendix A). At Atlixco (Figure 2C), the average number of weevils per plant was 10-fold the number in Teotihuacán (Figure 2D). This difference is related to fruit production (see below). Plants from the native Ticumán population had the lowest infestation by *T. soror* at both localities (Figure 2). There was also a significant effect of the range on the number of *Sphenarium purpurascens* (generalist herbivore) in Atlixco (Figure 2E) and on the number of *L. daturaphila* (specialist herbivore) in Teotihuacán (Figure 2F; Appendix A). For both herbivores, the native populations had more insects per plant. At the Teotihuacán site, we detected genetic variance (i.e., differences among families) in infestation by *T. soror* and *L. daturaphila* (Figure 2D,F).

Finally, in the Atlixco site, adults of *L. daturaphila* beetles were found on only four plants and no larvae were detected. On the same site, the tobacco sphinx moth, *Manduca sexta,* was found on two plants, and it was absent at Teotihuacán. 

### 2.2. Plant Height

Plants grown at Atlixco attained larger heights than plants grown at Teotihuacán (Figure 3A,B; Appendix A). In Atlixco, significant differences among populations and families were detected, and there was a marginally significant difference between ranges (Figure 3A). Here, the native plants (Mexico) were taller than non-native plants (Spain) (Figure 3A), and the plants from the native population of Teotihuacán were the tallest. 

At the Teotihuacán locality, plant height did not differ among plant populations (Figure 3B), and the plant height was generally small. 

### 2.3. Leaf Damage

The average proportion of leaf area lost to herbivores per plant was lower at Atlixco than at Teotihuacán (ca. 0.06 vs. 0.19, respectively) (Figure 3C,D; Appendix A). The non-native populations had slightly more herbivore damage than native populations (Figure 3C), and the (native) Ticumán population had the least damage. 

At the Teotihuacán locality, there was a significant difference in the proportion of leaf damage between ranges (Figure 3D). The plants from non-native populations had significantly less damage than the plants from native populations (Figure 2D; Appendix A).

### 2.4. Fruit Production

At the Atlixco site, fruit production differed significantly among populations (Figure 3E; Appendix A). Plants from the native Ticumán population produced, on average, more fruits than those from the Teotihuacán population, while the two non-native populations had intermediate values. 

At the Teotihuacán site, there were no significant differences in fruit production at the range, population, or family level (Figure 3F; Appendix A). The number of fruits per plant differed greatly between localities.

### 2.5. Alkaloid Production

Alkaloid concentrations were only measured at the Teotihuacán site. Plants from the non-native Zubia population produced the highest concentration of atropine, while plants from the native Ticumán population and the non-native Valdeflores population had the lowest atropine levels (Figure 4A; Appendix A). Scopolamine levels differed among populations and families (Figure 4B; Appendix A). The native Ticumán population had the highest mean scopolamine value, which differed significantly from the native Teotihuacán and the non-native Valdeflores populations (Figure 4B).

### 2.6. Accounting for Plant Damage

In the locality of Atlixco, regression analysis of plant damage as a function of plant height and the number of *S. purpurascens* failed to detect any significant effect on native or non-native plants (scopolamine concentration was not included in the model, as it was not quantified for the Atlixco site). 

In Teotihuacán, in contrast, damage in non-native plants was positively related to plant height and negatively related to scopolamine concentration (Table 1). The negative relationship between scopolamine and plant damage was detected for the native plants (Table 1). Atropine concentration and the number of larvae of *L. daturaphila* per plant were not related to plant damage.

### 2.7. Host Plant Selection for Oviposition

In the experiment at Teotihuacán, the number of *L. daturaphila* egg clusters laid on a plant was positively related to the number of *L. daturaphila* adults and negatively related to scopolamine concentration (Table 2; Appendix A). A covariance analysis of the number of egg clusters oviposited in relation to the plant’s range and the covariates scopolamine and adults of *L. daturaphila* did not detect differences between ranges. Both covariates—scopolamine and the number of adults—were significant (Table 2), but the interaction between range and the covariates and interaction between covariates were not significant. 

### 2.8. Natural Selection

Analysis of selection on plant size and herbivore damage, traits linked to plant fitness, indicated that these relate differently to plant fitness between the two experimental sites and between native and non-native plants. At Atlixco, native plants were selected to reduce damage by herbivores, whereas non-native plants were selected to increase plant height. No other selection gradient was significant at this site (Table 3). At Teotihuacán, the selection to increase plant size was significant for native and non-native plants (Table 3), while there was no selection on plant damage (i.e., resistance) (Table 3). 

## 3. Discussion

The levels of resistance to herbivores of non-native plants of *Datura stramonium* from Spain when returned to native environments in Mexico, were dependent on the ecological context (see [7]), namely the herbivores species present. The proportion of leaf damage to plants at the site where the generalist herbivore predominated (Atlixco) was lower than in Teotihuacán, and non-native plants were more severely damaged. In contrast, in the site where the specialist *L. daturaphila* is common (Teotihuacán), the native populations had more damage than non-native plants, and plant performance (i.e., size and fruit production) was no better for the non-native plants than the native plants. Finally, although there were no differences between ranges in the concentration of tropane alkaloids, the native population of Ticumán had the highest concentration of scopolamine, the alkaloid linked to plant defense in *D. stramonium*. Together, our experiments suggest that the reduction of selection on plant resistance in the non-native range of *D. stramonium* has not yet promoted strong divergence in plant defense against herbivores. 

### 3.1. Differences between Sites

The general vigor of the plants differed markedly between experimental sites. Plants grown at Atlixco attained larger sizes and higher fruit production than at Teotihuacán. These differences between sites can be related to climate: Atlixco receives more annual precipitation than Teotihuacán, and it also has less extreme temperatures. The sites also differ in species and abundance of herbivores (see below), probably also linked to climate. The experimental plot at Atlixco is set amid cultivars of sweet potato and ornamental (flower) production. In this site, soils contain fertilizer from cultivation of previous years. In contrast, at Teotihuacán, the experimental plot is an old field (abandoned 23 years ago), far from any currently cultivated lands. This locality is arid, and plant growth and reproduction are highly limited by climate as well as by herbivores. In this locality, interannual variability in the abundance of herbivores results in differential selection on resistance and plant relative growth rate over time [26]. Specifically, selection on resistance depends upon the abundance of *L. daturaphila* [26]. 

### 3.2. Herbivores

Previous studies of interactions between *D. stramonium* and its herbivores have found that the community of herbivores varies among localities in the native range [21,27] and in the non-native range [24]. In the current study, the generalist *S. purpurascens* was the main herbivore at Atlixco, and non-native populations had significantly more damage, perhaps due to lower levels of defense. In contrast, most leaf damage to plants at Teotihuacán was produced by the specialist beetle *L. daturaphila,* which is adapted to cope with alkaloids [27]. 

The diversity of herbivores between different ecological communities is thought to promote divergence in plant defense strategies (i.e., resistance and tolerance), driving populations to different trait values across space [20,28]. Studies of defense characters in *D. stramonium* have documented wide variation between populations in leaf damage, leaf trichome density, and concentration of two main alkaloids—atropine and scopolamine [19]. Moreover, contrasting selection on the concentration of atropine and scopolamine has been shown to occur depending on the dominant herbivore in a population (i.e., *E. parvula*, *S. purpurascens* or *L. daturaphila*) [21].

On the other hand, the incidence of seed predation by *T. soror* in populations of *D. stramonium* is highly variable, but evidence strongly suggests that the higher the population’s average scopolamine concentration, the lower the level of infestation by *T. soror* [22,27]. Our results can be interpreted in this direction. In both experimental sites, the population least infested by *T. soror* was Ticumán, which has the highest average scopolamine concentration. Hence, this system is a good candidate for assessing the evolution of defense in native and non-native populations driven by a specialized herbivore with direct effects on plant fitness. However, there was no general effect of native versus non-native origin on the incidence of seed predators in the native range; rather, the differences occurred at the level of individual populations. Furthermore, the differences in plant vigor may affect the amount of damage as well as fruit production, which in turn affects infestation by *Trichobaris*. Thus, ecological differences between localities constitute the material for geographic mosaics of selection [28]. 

### 3.3. Selection of the Plant for Oviposition 

*Lema daturaphila* is the main herbivore of *D. stramonium*, and larval load imposes selection on resistance [26,27]. Our results showed that the number of egg clusters oviposited on plants of *D. stramonium* is affected by the number of adults of *Lema daturaphila* and negatively by the concentration of scopolamine at the Teotihuacán site. Since *L. daturaphila* is a specialist and the coevolved herbivore of *D. stramonium*, it was not expected to be deterred by scopolamine [27]. 

The differences in the load of adult herbivores between plants of the two ranges (Figure 2F) are expected to account for differences in herbivory by larvae. Indeed, native plants had, on average, more damage than non-native plants (cf. Figure 3D). The reason for this preference for native plants is unknown, but it may be related to differences in the constitutive and/or induced defense between ranges. For instance, native and non-native plants might differ in the type/concentration of metabolites used as cues (e.gr. VOCs; [29]) by *L. daturaphila* to locate its host plants or the insect may be locally adapted to cope with the metabolites of its host plant. These two explanations are not mutually exclusive.

On the other hand, average values of egg clusters and larvae were not statistically different between plants of the two ranges (Appendix A). This suggests differential mortality, predation, or parasitoidism between plants of different ranges. Recent studies indicate that the eggs of *L. daturaphila* are readily parasitized by the wasp *Emersonella lemae* (Eulophidae) in most populations sampled in central Mexico, including the experimental site Teotihuacán (C.E. Villanueva-Hernández, Pers. Comm, October 2023). Likewise, parasitoidism of larvae by dipterans can account for up to 50% of larval mortality. Thus, it is likely that parasitoidism may contribute to differences in the average damage to plants [30]. Still, the causes of differential parasitoidism/predation between plants from native versus non-native ranges have not yet been experimentally determined [30]. 

Plants’ nutritional status and defense can also contribute to the amount of herbivory. Insects that feed on nutritionally stressed plant tissues may convert ingested tissue into growth and reproduction less efficiently [29], or their development times may be altered [31], thus changing the leaf area they remove from plants. At the same time, different times spent by larvae on plants may translate into different likelihoods of being parasitized. For instance, parasitoidism of *Manduca sexta* by a tachinid fly (*Drino rhoeo*) reduced the total leaf area consumed in *Datura wrightii* and variable assimilation efficiency of larvae [32].

Finally, recent results demonstrated differences in resistance to *L. daturaphila* larvae among F2 plants of *D. stramonium* derived from the cross of a local plant from Teotihuacán with a plant from the resistant population of Ticumán; this variation was mediated by an azaridone-like triterpenoid [27]. Thus, genetic/phenotypic divergence between native and non-native populations of *D. stramonium* cannot be ruled out. For instance, it is likely that other toxins that may be absent in populations that lack coevolved herbivores [14] explain the differential host recognition and, hence, herbivore load on plants [33]. This suggests that compounds like HIVPs and VOCs should be compared between native and non-native plants. 

### 3.4. Why Populations Are Differentially Damaged by Herbivores?

A previous study has shown that plants of *D. stramonium* in the introduced range (Spain) have much lower herbivore damage than native populations of the species in Mexico [24]. This evidence pointed to the lack of specialized herbivores in the introduced range. Plants in Spain are not immune to damage, but it is very low. In Mexico, plants receive much more damage, and it is frequent to find plants that have lost the entirety of their leaf area, fail to produce seeds, or are even killed by *L. daturaphila*, especially during outbreaks.

We expected that plants from Spain, when exposed to the coevolved specialized herbivores of *D. stramonium*, would suffer more damage than the native plants, based on the hypothesis that under isolation and in the absence of herbivores, non-native plants of *D. stramonium* would evolve toward lower levels of resistance and defense. In other words, costly defense (whether chemical or physical) is expected to be selected against [10,12] or shift [14,34]. The experimental evidence presented here indicates that non-native plants possess at least comparable levels of resistance to Mexican plants, expressing equivalent levels of the defensive chemicals atropine and scopolamine. In the following paragraphs, we offer some potential explanations and ideas to pursue in future research: (1)Native populations of *D. stramonium* differ in resistance characters depending on the species of herbivores present [20,21]. The defensive role of an alkaloid may therefore vary among locations of native populations of *D. stramonium* due to the variable herbivore community [27]. Within localities, the defensive role of a specialized metabolite can affect different herbivores differently (e.g., specialists and generalists). For instance, differential damage due to specialist herbivores on native and non-native congeners of *Senecio* seems to be related to the absence of some pyrrolizidine alkaloids in the latter, which are damaged mostly by generalist herbivores [33]. Chemical defense in the non-native populations of *D. stramonium* may thus function to prevent damage by generalist herbivores (present in their non-native range), resulting in low divergence from native populations in chemical defense (see [14]).(2)Native and non-native populations of *D. stramonium* may differ in the production of cues or ‘attractants’ to herbivores or herbivores’ natural enemies (or both), thus producing differences in herbivore loads between plants of different ranges. For instance, although plants that are naïve to specialized herbivores (i.e., non-native) may be preferred as host plants, herbivores’ natural enemies may also be attracted, counteracting the increased herbivore load. The eggs and larvae of *L. daturaphila* are attacked by parasitoids (wasps and dipterans, respectively). Experiments with volatile organic compounds (VOCs) of damaged and intact plants of *D. stramonium* from both ranges would help to determine whether these affect the level of parasitoidism and whether they are involved in parasitoid attraction [35]. Preliminary results indicate that damaged native plants of *D. stramonium* emit more VOCs than non-native plants [S. Velázquez-Márquez, Pers. Comm, October 2024].(3)The absence of specialist herbivores in the non-native range predicts the reduction in the production of alkaloids or other constitutive and induced specialized defensive metabolites by plants. Retention of these traits may occur because (i) they might still offer benefits to the plants, according to the type of herbivore [14]; (ii) their production costs are low, especially in environments with high resource availability, or (iii) these remain due to random genetic drift. Hypothesis (i) is likely, given that plants in the non-native range can encounter many generalist herbivores [14]. Furthermore, evidence suggests diminishing benefits of ER with increasing time since the hypothetical introduction [8]; a long period since introduction predicts an increase in the diversity of herbivores and more effects on performance. Because *D. stramonium* was introduced to Spain a long time ago, the low level of damage suggests that its specialized metabolites still function as defenses.The assumption that (ii) alkaloids are low-cost is not supported, but the increase in plant performance in rich environments can mitigate their cost. In this sense, *D. stramonium* in Spain grows in environments that are water- and nitrogen-rich compared to their habitats in Mexico [36]. Furthermore, *D. stramonium* retains the phenology displayed in the native range (summer annual), whereas most Mediterranean plants grow in winter; the adaptation of non-natives to new climates has been shown to affect their interactions with other species and can enhance their ability to invade [37]. Thus, three non-exclusive ecological contexts co-occur: asynchrony with potential plant competitors, high resource availability, and partial enemy release [7]. Finally, the retention of defenses due to random genetic drift is very likely, but this scenario predicts no genetic variation and further defense evolution.(4)Native and non-native populations of *D. stramonium* differ in chemical defense, and these might be linked to differences in abiotic factors (soil, fertility, climatic regime, etc.). For instance, the concentration of tropane alkaloids in *D. stramonium* varies genetically (cf. Figure 3B), but also in relation to the nitrogen level in the soil, temperature, water, and light incidence [38,39]. Thus, experimental studies must assess whether plants of *D. stramonium* from the native and non-native populations show differential expression of specialized (secondary) chemistry with changes in a given environmental factor (i.e., adaptive plasticity), thus increasing, or at least maintaining, fitness [40]. Experimental studies that control soil fertility, water availability, and/or microbiome will help to understand the evolution of non-native plants in a new range and the role of enemy release in the evolution of plant resistance/defense.

## 4. Materials and Methods

### 4.1. Study System

*Datura stramonium* (Solanaceae) is a member of a small genus of North American origin [41,42]. Currently, *D. stramonium* is widely distributed around the world [43]. This species is a robust summer annual plant that grows at disturbed sites where the natural vegetation has been removed. It grows in open, sunny sites along roadsides or streams and as a ruderal herb near current and former crop fields. In Mexico, it is common in temperate regions along the TransMexican Volcanic Belt, but it also occurs in localities with tropical climates [44]. In Spain, *D. stramonium* is considered to display “manifest invasive behavior” [45], occurring as a ruderal herb in anthropic habitats like the margins of rivers and streams, ditches, tailings, and other frequently disturbed habitats (see review [36]). Cultural knowledge by ancient Mexicans on medicinal plants, including different species of *Datura*, was compiled by Spanish naturalists and Friars [46,47]. In Spain, it was introduced from Mexico in the XVI century [13].

### 4.2. Herbivorous Insects

In Mexico, during the growing season (June–November), plants of *D. stramonium* are eaten by several insect species (see [15,27]). Commonly, *D. stramonium* is preyed upon by two leaf beetles (Coleoptera: Chrysomelidae), the tobacco flea beetle (*Epitrix parvula*) and the striped datura beetle *(Lema daturaphila*) ([48,49], by the seed predator (*Trichobaris soror*, Coleoptera: Curculionidae) [22,44,50], the tobacco sphinx moth (*Manduca sexta*; Sphingidae) [48] and by the generalist grasshopper, popularly known as “chapulín” (*Sphenarium purpurascens*, Orthoptera: Pyrgomorphidae) [15] (Figure 1). 

### 4.3. Sampled Populations

Two populations of *D. stramonium* from the native (Mexico) and two from the non-native (Spain) ranges were selected to be grown and exposed to the herbivorous insects in the native range (Mexico). Because Mexican populations of *D. stramonium* vary in herbivore communities, we selected two previously studied populations—Teotihuacán and Ticumán—that differ in damage by herbivores and alkaloid production [18,19,26,51]. In the case of the non-native range, most previously field-studied populations are from southern Spain (Andalusia); thus, we selected two populations at random for the experiments. From the 14 previously studied populations, we selected the population Valdeflores (53 km NW of the City of Seville) and the population La Zubia (henceforth Zubia; 5 km south of the City of Granada) [24]. In both localities, the proportion of leaf area removed by herbivores was very low (0.02 and 0.034, respectively) [24].

### 4.4. Experimental Localities 

We established two experimental common gardens at two locations in central Mexico—Atlixco, in the State of Puebla (2017) and Teotihuacán, in the State of Mexico (2019) (Appendix A). These localities are separated by ca. 100 km linearly and differ in climate (Teotihuacán, Cwb, while Atlixco, Cwa; Appendix A). They belong to the Basins of Puebla and Mexico, respectively, separated by the Sierra Nevada. The populations of *D. stramonium* from the non-native range were collected in the region of Andalusia, which has a typical Mediterranean climate (Csa) (Appendix A). 

### 4.5. Plant Material

In each range (Mexico or Spain), seeds from different maternal plants in each population were sampled. From every plant collected per population, we took a single fruit (natural progenies; [52]). Fruits were bagged individually and labeled. In the lab, a sample of seeds from each maternal plant (*n* = 50) was surface disinfected with sodium hypochlorite solution (1.25%) for 5 min and then rinsed with distilled water [53]. To improve germination, we removed the seed coat using a fine-tip tweezer. The seeds of each plant were settled in a Petri dish lined with moistened filter paper and incubated in a plant growth chamber (Conviron, Inc., Winnipeg, MB, Canada) at 35 °C and 25 °C, and 14 and 10 h of light:dark (day—night), respectively. Seedling emergence was checked daily. Generally, germination took place within a week.

Emerged seedlings were transferred to 250 mL pots filled with a sterilized mixture of sand and perlite (4:1) and placed on benches in a glasshouse. Once the plants’ first pair of true leaves had expanded, the plants were transported to the experimental site for planting. 

### 4.6. Experimental Design

Transplantation followed a completely random design [54]. In each experimental site, plants were randomly allocated to a position in the plot, given by a column and a row, at equidistant positions 1 m apart. In the field, we previously dug a hole in the soil for each plant. In both sites, supplemental watering was used to allow plant establishment. At the Teotihuacán site, where rain is very scarce, the plants were watered every 3–4 days for the first month. In Atlixco, where there is higher natural precipitation, plants were watered three times (near the beginning, midpoint, and end of the season) in addition to natural rainfall. Weeds were removed manually every two weeks in both sites.

### 4.7. Data Collection

Plants were monitored to record establishment and colonization by herbivores. *Epitrix parvula* arrives early in the season, colonizing young plants [15,20], and all other herbivores arrive later. At Teotihuacán, when *L. daturaphila* colonized the plants, we counted the number of adults, the number of egg clusters oviposited and, a week later, the number of larvae groups on each plant. At the same time, we obtained a nondestructive estimation of the proportion of leaf damage performed visually by a single observer. 

*Sphenarium purpurascens* colonizes the plants at later stages in the season since it passes through different nymphal stages and reaches adulthood late in the summer [55]. This flightless generalist grasshopper consumes *D. stramonium* only if there are no other available food plants [15]. This insect was common in the crops at Atlixco.

At reproduction but before plant senescence, we measured plant size (height) and collected a sample of leaves to estimate the leaf area removed by herbivores (see below). At the same time, we collected a sample of leaves, usually 2–3, to estimate the concentration of the tropane alkaloids atropine and scopolamine in the lab. Using gloves, we collected the leaves, rinsed them with distilled water, wrapped them in aluminum foil, labeled them, and immediately deposited them in liquid nitrogen. In the lab, these samples were transferred to an Ultra freezer (Thermo Fisher Scientific, Waltham, MA, USA) until analysis (see below).

Fruits produced by each plant were collected individually and labeled. In the lab, they were open to record infestation by the weevil *T. soror*. We counted the number of adults and larvae in each fruit (see [22]). 

### 4.8. Plant Resistance to Herbivores

To evaluate general plant resistance, a random sample of leaves (*n* = 20–30) per plant was collected, pressed, labeled, and dried to obtain the proportion of leaf area damaged more precisely. When a plant had fewer than 20 leaves, we collected most of the leaves of the plant. In the laboratory, we measured the total leaf area of each leaf as well as the damaged area using the WinFolia software (WinFOLIA™, Regent Instruments Inc., Quebec, QC, Canada; https://regent.qc.ca/assets/winfolia_about.html). 

We obtained an estimator of plant resistance (*R*_i_) per plant as follows: $$R_{i}=1-(\sum_{1}^{n}\frac{{LAD}_{i}}{{TLA}_{i}})$$
where *LAD_i_* and *TLA_i_* are the leaf area damaged and total leaf area per plant, respectively.

### 4.9. Tropane Alkaloids of Datura stramonium 

The concentration of the two main alkaloids produced by *D. stramonium*, atropine and scopolamine, was determined only for plants from the Teotihuacán site. We collected leaf tissue at plant reproduction to extract the leaf alkaloids. To do this, leaf tissue was freeze-dried for 72-h and ground in a mortar. 100 mg of powdered tissue was suspended in 1 mL of 80% methanol and 1% formic acid in a 2 mL Eppendorf-type tube. Then, leaf samples were shaken using a TissueLyser II (Qiagen, Germantown, MD, USA) for 45 s at 30 Hz, followed by sonication for 15 min, and centrifugation at 14,000 rpm for 20 min at 4 °C. Finally, 700 µL of the supernatant was taken and placed in an Agilent glass vial for gas and stored at 4 °C until analysis by Agilent 6230 time-of-flight LC/MS system (High-Performance Liquid Chromatography with Time-of-Flight Mass Spectrometry; Agilent, Santa Clara, CA, USA).

Chromatographic separation was carried out on a column that was cleaned before injection, as described in previous studies [27,51]. For the sample injection, they were first made up to 1 mL by adding 300 µL of MeOH. Chromatography was performed on an Agilent 1260 Infinity system coupled to an LC/MS-6230 time-of-flight (TOF) accurate mass spectrometer using an Agilent Technology 1200 Infinity autosampler. The Agilent ZORBAX HPL column; Agilent, Santa Clara, CA, USA) was used for metabolite separation. The chromatographic conditions for the samples were as follows: 0.00 min with 90% A eluent, 10 min with 10% A eluent, 17 min with 90% A eluent, and 17.10 min with 90% A eluent. These chromatographic conditions were maintained for 5 min to equilibrate the column. The total run time was 23 min per sample, with the column temperature set at 50 °C. The samples were kept at 4 °C during the injection process, and 1 µL was taken for injection per sample [27,51].

The quantification and detection of alkaloids were performed using two standards, atropine and scopolamine (Sigma-Aldrich, St. Louis, MO, USA). A dilution of 1:100 mg/mL was prepared, and volumes of 2, 4, and 8 µL were injected to create the calibration curve. The concentration of alkaloids (µg/mL) in a sample was calculated using the standard curve generated from the dilutions of each alkaloid. Analyses were carried out at the National Laboratory of Health, Faculty of Higher Studies Iztacala, UNAM.

### 4.10. Statistical Analyses

The experimental design allows the partition of phenotypic variance due to range, populations (within range) and families (within populations). This last term, in turn, allows the detection of broad-sense genetic variance [35]. The number of families planted per population in each site is given in Appendix A. Because of the hierarchical nature of this design, we tested the effects range (Spain and Mexico) of populations within range and families within population on response variables by means of nested analyses of variance (ANOVA). Predictor variables were declared as random effects. Prior to analyses, variables were transformed to improve normality as follows: the proportion of leaf damage exerted by herbivores was arcsin (*x*) transformed, whereas counts were log (*x* + 1) transformed. The quantity of tropane alkaloids, atropine and scopolamine, were log-transformed. The analyses were performed on transformed values, whereas graphs were constructed using actual values, for comparison. 

#### 4.10.1. Host Plant Selection for Oviposition

Females of *Lema daturaphila* select plants to oviposit; we counted the number of adults, the actual number of egg clusters laid and the number of larvae on each plant. We assessed whether the number of egg clusters (actual selection of a plant as host) is affected by the number of adults of *L. daturaphila* and plants’ atropine and scopolamine concentration in leaves by means of multiple linear regression. A covariance analysis assessed whether the relationships of these covariates differed among populations.

#### 4.10.2. Natural Selection 

We evaluated whether selection acted on plant size (height), a major determinant of plant fitness in *D. stramonium* [15], and damage by herbivores. The number of fruits produced by each plant was recorded and used as an estimator of maternal fitness. Plant relative fitness (*w*_i_) was obtained by dividing each individual plant value (no. of fruits) by the population average for the same variable. Individual fitness was regressed on plant size and infestation by herbivores in each population. Variables were standardized before analyses; selection gradients (*β_i_*) of plant traits are thus in standard deviation units [56]. In the experiment at Teotihuacán, we also assessed whether alkaloids affect fitness. Selection analysis was performed between ranges (native and non-native) because the low statistical power with the sample sizes of each population precluded their independent analysis. 

All analyses were performed in JMP^®^ 9.0.0 (SAS, Institute, Inc., Cary, NC, USA, 2010). 

## 5. Conclusions

The results presented here support the idea that the low levels of damage in populations of *D. stramonium* in its new range in Spain are a consequence of enemy release. However, results of experiments in two different sites in the native range and in the presence of native herbivores demonstrated that strong divergence in plant defense (alkaloids) and plant resistance to herbivores between ranges have not occurred, at least for the traits analyzed. Variations in plant damage, alkaloid production, and plant performance (fitness) are attributed mainly to differences between populations. However, to a lesser extent, there were differences between ranges, as in the case of damage by the generalist grasshopper that preyed upon non-native plants, while the specialist *L. daturaphila* caused more damage to the native plants. Genetic variation for resistance, alkaloid production, and infestation by herbivores remain in plant populations of *D. stramonium,* thus enabling further evolution. Our results underscore the need for studies that assess the relationship between specialized plant metabolites, herbivores, and herbivores’ natural enemies in native and non-native *Datura* populations.

## Figures and Tables

**Figure 1 plants-13-00131-f001:**
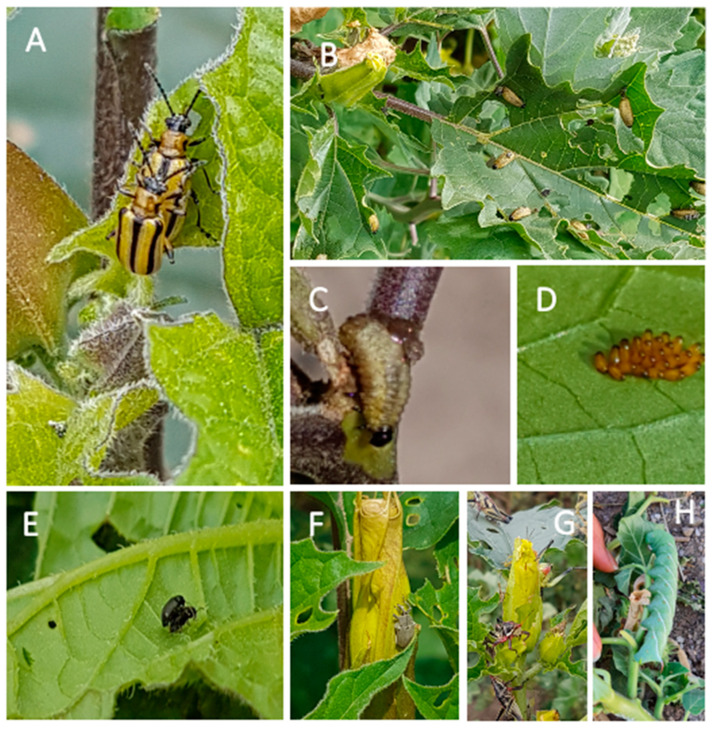
Herbivores of *Datura stramonium*. (**A**) Adults, (**B**) larvae, (**C**) larvae eating the stem, and (**D**) egg cluster of *Lema daturaphila*. (**E**) *Epitrix parvula*; (**F**) *Trichobaris soror*; (**G**) *Sphenarium purpurascens*, and (**H**) *Manduca sexta*. Herbivores in panels (**A**–**F**) were present at both Teotihuacán and Atlixco (see Figure 2). Only four plants had *L. daturaphila* at Atlixco, while *S. purpurascens* was absent at Teotihuacán during our sampling dates. *Manduca sexta* (**H**) was only found at Atlixco.

**Figure 2 plants-13-00131-f002:**
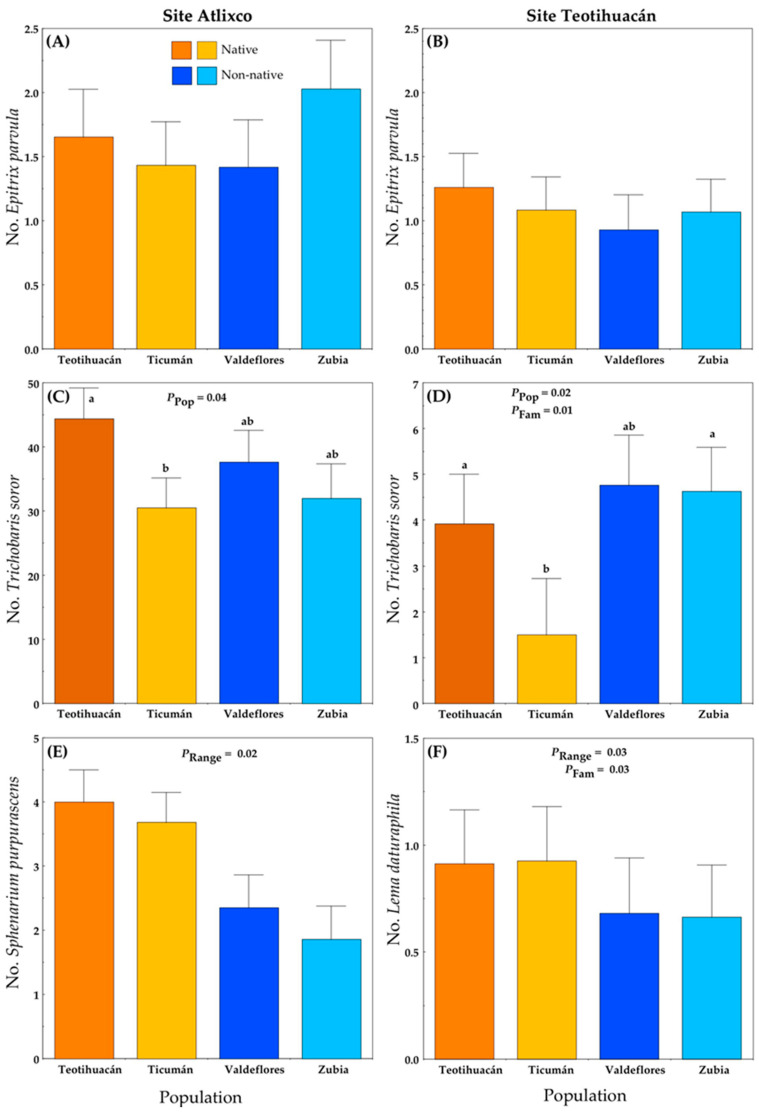
Mean number (+S.E.) of herbivores on native and non-native plants of *Datura stramonium* at two experimental sites: Atlixco (**A**,**C**,**E**) and Teotihuacán (**B**,**D**,**F**). Significant effects are indicated, and different letters within a panel indicate significant differences between populations (*p* ≤ 0.05) according to post hoc *t*-tests (see also Appendix A). Range = Between ranges (Mexico/native, Spain/non-native); Pop = between populations within range; Fam = among families within population.

**Figure 3 plants-13-00131-f003:**
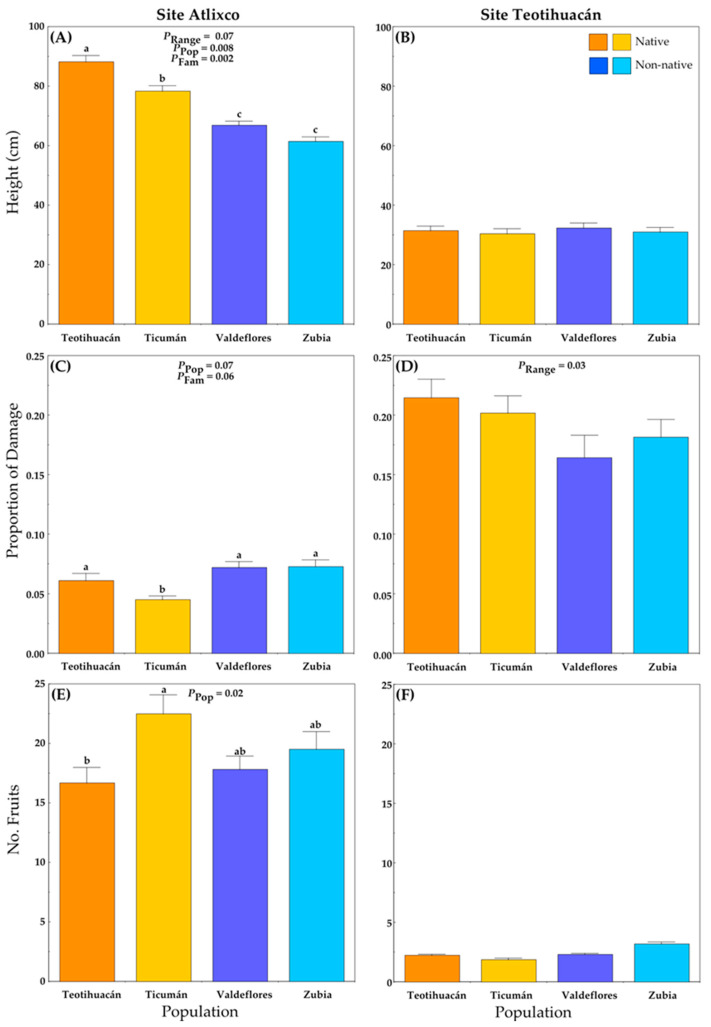
Average values (+S.E.) of plant traits of native and non-native plants of *Datura stramonium* at two experimental sites: Atlixco (**A**,**C**,**E**) and Teotihuacán (**B**,**D**,**F**). Significant effects are indicated, and different letters within a panel indicate significant differences between populations (*p* ≤ 0.05) according to post hoc *t*-tests (see also Appendix A). Range = Between ranges (Mexico/native, Spain/non-native); Pop = between populations within range; Fam = among families within population.

**Figure 4 plants-13-00131-f004:**
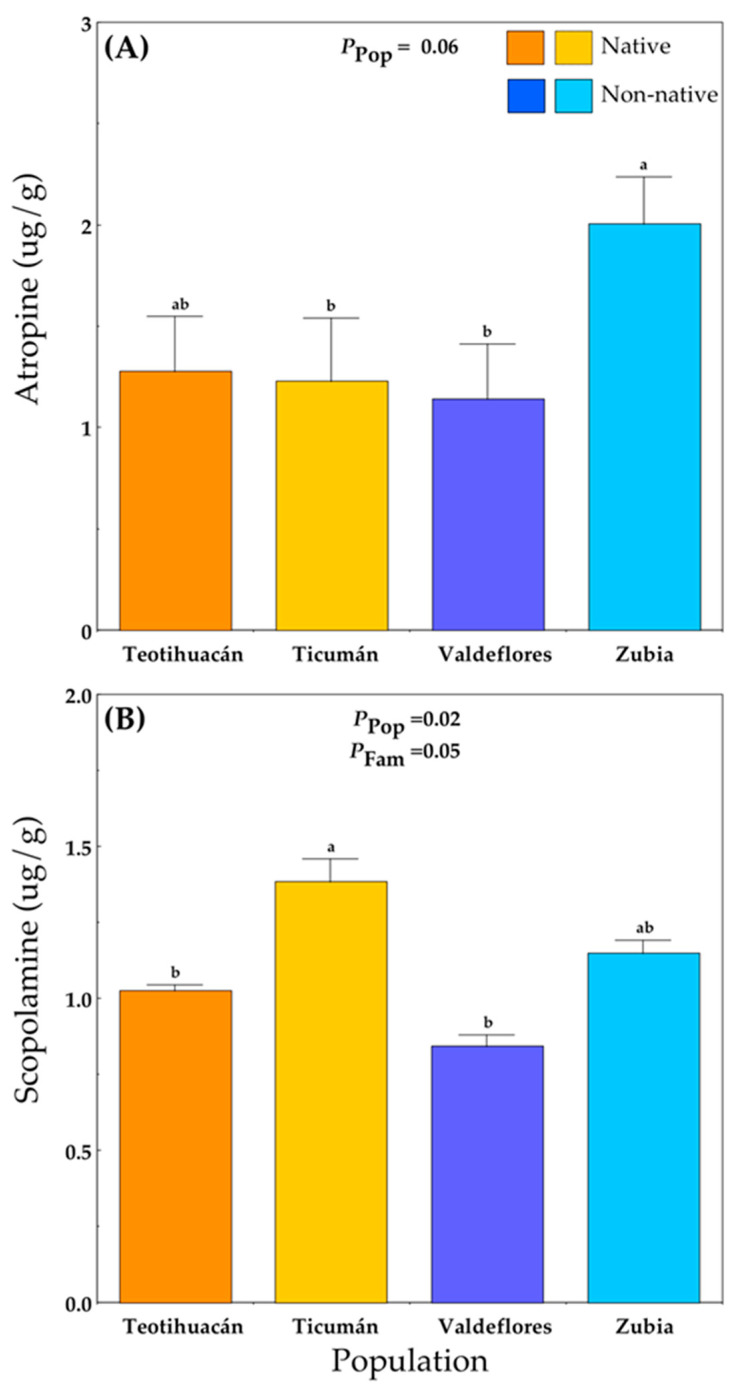
Mean (+S.E.) atropine (**A**) and scopolamine (**B**) concentration in native and non-native plants of *Datura stramonium* at the experimental site Teotihuacán (see Appendix A). Significant effects are indicated, and different letters within a panel indicate significant differences between populations (*p* ≤ 0.05) according to post hoc *t*-tests (see also Appendix A). Pop = between populations within Range; Fam = among families within population.

**Table 1 plants-13-00131-t001:** Regression analysis of damage by herbivores of native and non-native plants of *Datura stramonium* at the site of Teotihuacán as a function of plant height, alkaloid concentration, and number of larvae of *Lema daturaphila*. Significant estimates are indicated by asterisks and bold text. N (natives) = 33, N (non-natives) = 37.

	Native Plants			Non-Native Plants		
Term	Estimate (Standard Error)	*t* Ratio	Prob > |*t*|	Estimate (Standard Error)	*t* Ratio	Prob > |*t*|
Intercept	**0.2681** (0.0676)	**3.96**	**0.0005 ***	**0.2157** (0.0452)	**4.77**	**<0.0001 ***
Plant Height	0.0023 (0.0018)	1.26	0.216	**0.0044** (0.0014)	**2.96**	**0.0057 ***
Atropine	0.0313 (0.03657)	0.86	0.398	0.0472 (0.0405)	1.16	0.2532
Scopolamine	**−0.0647** (0.0317)	**−2.04**	**0.050 ***	**−0.1155** (0.0349)	**−3.31**	**0.0023 ***
No. *Lema* larvae	0.0526 (0.0326)	1.61	0.118	0.0426 (0.0292)	1.46	0.1549

**Table 2 plants-13-00131-t002:** Analysis of covariance of the number of egg clusters deposited on plants of *Datura stramonium* between ranges (native/Mexico and non-natives/Spain). The number of adults of *L. daturaphila* and the plant’s scopolamine concentration were the covariates. None of the interactions was significant (*n* = 170). The most parsimonious model (i.e., the model with the lowest AIC score) is presented. * *p* < 0.05.

Term	Estimate	S.E.	*t* Ratio	Prob > |*t*|
Intercept	1.3846	0.1494	9.27	<0.0001 *
Range	−0.0166	0.0636	−0.26	0.7936
**Adults of *Lema daturaphila***	0.7426	0.1055	7.04	<0.0001 *
**Scopolamine**	−0.2616	0.5653	−2.32	0.0214 *
**Effect Tests**				
**Source**	**Sum of Squares**	**d.f.**	***F* Ratio**	**Prob > *F***
Range	0.0456	1	0.0687	0.7936
**Adults of *Lema daturaphila***	32.9161	1	49.5296	<0.0001 *
**Scopolamine**	3.5847	1	5.3940	0.0214 *
Error	110.3195	166		
Total	155.9339	169		

**Table 3 plants-13-00131-t003:** Selection gradients (*β*) on plant height and damage by herbivores on native and non-native plants of *Datura stramonium* grown at the Teotihuacán site. The standard error of the estimate is given in parentheses. * *p* < 0.05; ** *p* < 0.001.

		Atlixco Site			Teotihuacán Site	
Range	*n*	Plant Height *β*	Plant Damage *β*	*n*	Plant Height *β*	Plant Damage *β*
Native	125	−0.0693 (0.0561)	−0.1479 ** (0.0561)	56	0.5006 ** (0.1646)	0.1102 (0.1578)
Non-native	108	0.1694 * (0.0703)	−0.0557 (0.0500)	63	0.6372 ** (0.1318)	−0.0521 (0.1431)

## Data Availability

The data presented in this study are available on request from the corresponding author.

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
