# Peer review of "A Trip Back Home: Resistance to Herbivores of Native and Non-Native Plant Populations of Datura stramonium"

_plants, 2024, doi:10.3390/plants13010131_

Round 1

Reviewer 1 Report

Comments and Suggestions for Authors

Manuscript ID: plants-2694878Title: A trip back home: Equal resistance to herbivores of native and non-native plant populations of Datura stramonium

Authors: Juan Núñez-Farfám *, et al.

Review

In the paper authors aimed to assess, if invasive (non-native) plants of D. stramonium from Spain differs from native plants from Mexico in (a) plant resistance, (b) in production of tropane alkaloids, and (c) in selectivity by natural herbivore of the plant to oviposition. To test their hypothesis – namely, that in the native habitat and in the presence of native, specialized herbivores, non-native populations will receive more damage and express lower defense than the natives – authors established two experimental sites, grow plants from the two origins (Mexico, Spain) and exposed them to natural herbivores of D. stramonium.

I fully agree with their idea and aim, however, manuscript in current form is not suitable for publication.

Some general remarks:

1.     It is commonly accepted, that common name of the species should be given on the first mention.

2.     Why authors use word “genotype” they did no genotyping. In the Table S2 they use “families”, and “progenies” – we think, the last one is correct; see 4.3. If we understand correctly, all experimental pool of plants originate from two maternal plants, one of these was from Mexico, the other from Spain?

3.     Please clearly describe your understanding of “origin”, “population” and  “genotype”, see Table S1, and Material and Methods, for any reader to understand what you mean,

4.     Material and methods must be described for reader to understand and be able to repeat the study. In the current form it is not understandable.

5.     It would be nice to see measurement units for all variables, as now “estimates” in the Table 3 for example, are not clear.

6.     I think there are mistakes in the calculations or in figures (see below).

7.     Based on the situation, it is highly recommendable to put all data on the Supplement file, possibly Excel, and give reader possibility to test at least differences. This does not mean accusation, maybe because of the manner of presentation some statements in the paper looks doubtful.

8.     Please use the same Y axis values for two sites (Figure 2)

9.     Dimensions must be shown for Y avis

10.  Manuscript do not conform to Template in many aspects.

11.  Beginning of the Page 6 is not readable

Title and keywords: use common name of the species in one of these

Abstract

Introductory part too long, result part too short.

Mistypes: Lines 26, 33, 35

Already from the abstract is not clear, what is relation between origin of a tested sample (Spain vs Mexico), populations (5 shown in the Table S2A, but only 4 in Figures 2, 3 and in the text).

Andalusia is in Spain, of course, but later in the paper Andalusia is not used.

Language in very far from clear one.

Introduction

First paragraph is almost solely based on [3], but there are many other sources to use in this context, e.g., Liu, H., & Stiling, P. (2006). Testing the enemy release hypothesis: a review and meta-analysis. Biological invasions8, 1535-1545.

Source [4] is not cited,

Journal is using citations by numbers, but see Lines 62, 66 in the text.

Results

Text on the composition of herbivores is not easy to understand. Just a proposal: can a caption of the figure 1 be presented in a way to show differences between two study sites?

Can herbivores also be presented by their common names on the first use?

Figure 2. There are either mistakes in calculations, or in presentation. Let me explain: when one has average and SE values in graphic form, overlapping and non-overlapping of SEM can have different interpretation, if sample size is different. See https://www.graphpad.com/support/faq/spanwhat-you-can-conclude-when-two-error-bars-overlap-or-dontspan/

However here, based on Table S2B, sample sizes are close. Therefore, it is not clear

·         What is “ab” in F part;

·         How can B or H part have “ns” if G or F have significant differences.

It should be nice to see, if authors tested normality of distribution of used variables, and what was the result.

Could different colors be used for each population in figures 2 and 3, and also to see native / non-native populations?

Line 123: By the way, there is no E part in the Figure 2 J

Line 129: authors use “leaf damage” here, “plant damage” in Figure 2, but “plant resistance” in chapter 4.4. Complete mess – please clarify. Line 135” percentage comes from nowhere, and really not from the mentioned Figure 2 or Table S1.

Chapter 2.3. Line 139: wrong, Figure 2B do not refer to Atlixco. Secondly, hard to believe there are no differences in Figure 2B, compared to Figure 2F.

Figure 2D and 2H – what is “Total number herbivores” on Y axis? How it compares to 4.6?

Chapter 2.4. – text is wrong, or Figure 2 D,H is wrong, as it refers ns for the both parts, but text says “more”.

Discussion

This part, 3.1 put a doubt on at least “location” variable. If you look to 4.2, there are much more important differences than “differ slightly in climate”.

Therefore, I see publication of this manuscript in Plants not possible in the current form.

Comments on the Quality of English Language

In many places, text is very difficult to understand.

Reviewer 2 Report

Comments and Suggestions for Authors

This study compares native and introduced populations of Datura stramonium while focusing on insect herbivory and plant defense traits. The findings of the research are intriguing, but there are a lot of issues that require substantial revisions: 

Major revision:

Introduction: The final paragraph mentions two scientific questions, namely, the differences in insect herbivory and plant defense levels between native and invasive populations. However, these two scientific questions have already been addressed in previous studies. Therefore, it's unclear where the novelty of the author's research lies. Additionally, the author brings up insect oviposition selection in the last paragraph, even though there is no prior discussion on this topic in the preceding paragraphs. Consequently, it's unclear why the author is focusing on insect oviposition selection.

Results: The statements in the first paragraph lack corresponding data to support them. The referenced Figure 1 is only a photograph, depicting the herbivorous insects of the species and their feeding symptoms, making it appear more like background information on the study site rather than results. Therefore, I suggest moving it to the Materials and Methods section. When presenting results, it's essential to include relevant statistical measures like p-values, F-values, and degrees of freedom rather than simply referring to Table XX. 

Discussion: The Discussion section seems to have too many points, making the content fragmented and lacking a clear focus. For instance, the differences in herbivorous insects between the two locations are not the main concerns of this paper and do not require extensive discussion. The emphasis should be placed on addressing the scientific questions raised in the Introduction. Furthermore, this paper has several limitations, including: 1) The selection of only two populations from the native and invasive regions, with no clear rationale for this choice; 2) Limited measurements of chemical defense traits and no direct evidence to establish a connection between these chemical defenses and the herbivorous insects of interest. Therefore, a section discussing the limitations of the paper should be included in the Discussion. In addition, there is a notable absence of discussions related to common topics in the field of evolutionary studies of defense in exotic plants. These include issues such as how the insect feeding specialization, types of plant chemical defense (qualitative vs. quantitative), abiotic factors (temperature, moisture, nutrients, latitude etc.), and plant defense strategies may impact the evolution of defense in exotic plants."

 Materials and method: Many critical pieces of information have been overlooked in the Materials and Methods section. In studies concerning the rapid evolution of exotic species, the selection of populations from the native and invasive regions is of utmost importance. It is advisable to supplement this with genetic evidence to demonstrate a genetic relationship between native and invasive populations. Even when such evidence is lacking, it's essential to thoroughly explain the basis for selecting these populations. However, the current paragraph does not provide any justification for the chosen populations. Moreover, the experimental procedures appear overly simplistic, with many details left unclear. For example, it is not specified under what conditions the plants were grown, whether they were directly planted in the field or placed in pots in the field. Was the exposure of plants to herbivory throughout their growth or limited to specific stages? Were measurements of plant size and leaf consumption conducted after harvesting the plant material or continuously throughout the plant's growth? How were these metrics recorded? These details are crucial, as they aid readers in assessing the reliability and comparability of the study's results.

Specific comments:

Lin 51~53: This sentence needs reference。

Line 55: The descriptions throughout the paper need to be consistent; they can't switch between 'non-native' and 'introduced'. It can be suggested to change as 'alien species and introduced range'.

Line 63: The formatting of the references is inconsistent, switching between numerical and 'author+year' styles.

Line 76: deletS, and thoroughly check the content.

Figure 2: The native and introduced populations should be marked as different color.

Line 129-136: When describing this part of the results, you can start by presenting the overall findings. Then, discuss whether the general trends observed in the experiments at both locations are consistent. If they are consistent, you can simply state that the results from Location XX align with the overall pattern. If they are not consistent, then address the specific discrepancies separately.

Line132: delete the redundant “Fig.and thoroughly check the content.

Figure 3: The native and introduced populations should be marked as different color.

Line 227: insect”。

Line 278: remove the.Between “main” and”herbivores”.

Line318-319: Besides the issues discussed below, there's another crucial aspect that the author has overlooked in their explanation: this study was conducted in the field, which means it cannot distinguish between constitutive and induced plant defenses. Therefore, even if there is no difference in overall resistance levels, different defense strategies may still vary between native and introduced populations.

Line322-325: I don't understand the meaning of this sentence. Is it trying to convey that chemical defense compounds can also attract the natural enemies of herbivorous insects?

Line332-337: This is a valid point, but there is existing research on this topic, and thus the absence of a reference to a specific paper here is not appropriate.

Line457-458: I don't understand the meaning of this sentence.

Comments on the Quality of English Language

This article contains numerous writing and formatting errors. For instance, Table 1 and Table 3 are identical, and Table 1 is repeated three times, with the second instance completely covering the text. The reference format in the Introduction is inconsistent, and there are spelling errors, such as "insect" in Line 227. Please urge the authors to thoroughly check the content and formatting to demonstrate a meticulous and careful approach to their work.

Reviewer 3 Report

Comments and Suggestions for Authors

Plants

Farfan et al.

A trip back home…

General

An interesting approach, but unfortunately, I get quite confused by reading this paper. Language should be improved, but mainly - the text should become clearer, more structured and maybe shorter. I also find the presentation of the results quite confusing.

Figure 2. Not feasible to use such a small font. The legend is insufficient, what is the meaning of, for ex "Atlixco" experiment - who cares about geographic locations, the type of experiment is important. Why were insects pooled (D), and not shown separately? Statistics - no difference vs difference in B and F ?

Tables. How useful is it to show details of the regression analysis in the Results (Supplement would be the place). Instead, a more explicit and detailed presentation of the results wouldn't hurt.

Overall, can't really review tyhe paper in its present shape.

Intro

First para - improve clarity?

79. 0.025 - what?

Overall: shorter & clearer

Results

139. "only", what else was looked at?

Minor

20. checks?

21. Avoid using 2 synonymous terms with hyphens

22-23. meaning?

37-42. Unclear if these conclusions properly reflect the results

Comments on the Quality of English Language

-

Author Response

Please see trhe attachment

Round 2

Reviewer 1 Report

Comments and Suggestions for Authors

 A trip back home: Equal resistance to herbivores of native and non-native plant populations of Datura stramonium

By Juan Núñez-Farfám *, Sabina Velázquez-Márquez, Jesús R. Torres-García, Ivan Mijail De-la-Cruz, Juan Arroyo, Pedro Luis Valverde, César Mateo Flores-Ortiz, Luis Barbo Hernández-Portilla, Diana Elizabeth López-Cobos, JAvier Daniel Matías

Review round 2

I appreciate authors work in answering my comments and revising manuscript text. In my opinion, it is suitable for publication in Plants after correcting some small deficiencies, asking for a minor revision.

General comments:

·         Use correct punctuation, [1–4], not [1, 2, 3, 4] (Line 52). Check referring throughout

·         Use common names  + Latin names for the first time

·         Why your figure are not in color? They are free in MDPI J

Abstract: in the journal website it is written “The abstract should be a total of about 200 words maximum” – this, please shorten your text. Otherwise, it is good.

Introduction

Line 59: mistype, missing comma in references

Lines 68, 71: correct parentheses to [ ]

Line 89: [15–20], see throughout

Results

Line 139: (Fig. 2F; Table S1A).

Line 188: missing ;

Material and Methods

Line 520: (Table S2 B) –check punctuation, formerly you do not use space

Back Matter

Supplementary materials: all supplementary materials must have captions, and these captions need to be presented here, see Template

References

Use correct dash length for page numbers, eg, Line 701: 429–453, not  429-453.

Lines 703, … , 801, 808, 812, 821, 828: volume in italics, please

Information for [56]:

The Measurement of Selection on Correlated Characters

Russell LandeStevan J. Arnold

Evolution, Vol. 37, No. 6 (Nov., 1983), pp. 1210-1226 (17 pages)

https://doi.org/10.2307/2408842

Author Response

We appreciated all suggestions provided by you to improve the manuscript.

Reviewer 2 Report

Comments and Suggestions for Authors

The auther have addressed all the concerns of mine, and I have no further comments

Author Response

Dear Reviewer,

Thank you for your feedback and comments. We have introduced the corrections in the new version of the manuscript and changed the figures.